# Bioinformatic Analysis Reveals the Role of Translation Elongation Efficiency Optimisation in the Evolution of *Ralstonia* Genus

**DOI:** 10.3390/biology12101338

**Published:** 2023-10-16

**Authors:** Aleksandra Y. Korenskaia, Yury G. Matushkin, Zakhar S. Mustafin, Sergey A. Lashin, Alexandra I. Klimenko

**Affiliations:** 1Systems Biology Department, Institute of Cytology and Genetics, Siberian Branch of the Russian Academy of Science, Lavrentiev Avenue 10, Novosibirsk 630090, Russia; korenskaia@bionet.nsc.ru (A.Y.K.); mustafinzs@bionet.nsc.ru (Z.S.M.);; 2Kurchatov Genomics Center, Institute of Cytology and Genetics, Siberian Branch of the Russian Academy of Science, Lavrentiev Avenue 10, Novosibirsk 630090, Russia; 3Department of Natural Sciences, Novosibirsk National Research State University, Pirogova St. 1, Novosibirsk 630090, Russia

**Keywords:** translation elongation efficiency, translation in prokaryotes, *Ralstonia*, evolution of *Ralstonia* genus, pathogenic bacteria

## Abstract

**Simple Summary:**

Bacteria belonging to the *Ralstonia* genus are of great interest for study since this genus includes both phytopathogenic species and soil opportunistic human pathogens, forming two phylogenetic branches. The difference in phenotype of related species is formed by differences in how proteins and other molecules bacteria synthesize, as well as the level of their synthesis. In this study, we investigate the evolution of the *Ralstonia* genus, focusing on characteristics of a particular stage of protein synthesis. This is a stage at which the protein sequence is synthesized, named translation elongation. The efficiency of this stage plays a role in the modulating of gene expression, while characteristics of this stage considered being more conservative allow us to use it for comparison at the genus level. We calculated the elongation efficiency based on genomic data and found that characteristics of translation elongation efficiency optimization diverge in accordance with the phylogeny of *Ralstonia*. Differences in these characteristics of genomes as a whole are also reflected in differences in the efficiency of translation elongation of individual genes, resulting in differences in sets of potentially highly expressed genes in soil and phytopathogenic bacteria. These results are valuable for understanding the evolution and adaptation of bacteria in different environments.

**Abstract:**

Translation efficiency modulates gene expression in prokaryotes. The comparative analysis of translation elongation efficiency characteristics of *Ralstonia* genus bacteria genomes revealed that these characteristics diverge in accordance with the phylogeny of *Ralstonia*. The first branch of this genus is a group of bacteria commonly found in moist environments such as soil and water that includes the species *R. mannitolilytica*, *R. insidiosa*, and *R. pickettii*, which are also described as nosocomial infection pathogens. In contrast, the second branch is plant pathogenic bacteria consisting of *R. solanacearum*, *R. pseudosolanacearum*, and *R. syzygii*. We found that the soil *Ralstonia* have a significantly lower number and energy of potential secondary structures in mRNA and an increased role of codon usage bias in the optimization of highly expressed genes’ translation elongation efficiency, not only compared to phytopathogenic *Ralstonia* but also to *Cupriavidus necator*, which is closely related to the *Ralstonia* genus. The observed alterations in translation elongation efficiency of orthologous genes are also reflected in the difference of potentially highly expressed gene’ sets’ content among *Ralstonia* branches with different lifestyles. Analysis of translation elongation efficiency characteristics can be considered a promising approach for studying complex mechanisms that determine the evolution and adaptation of bacteria in various environments.

## 1. Introduction

*Ralstonia* is a bacterial genus with a complicated systematic history. Currently, it is regarded to be comprised of two branches with different lifestyles [1,2]. The first is a group of bacteria commonly found in moist environments that includes species *R. mannitolilytica*, *R. insidiosa*, and *R. pickettii*, which are also described as nosocomial infection pathogens [1,3,4]. The second group is plant pathogenic bacteria with a broad host range, including such agricultural plants as potato, tomato, eggplant, ginger, etc. [5]. This group consists of the species *R. solanacearum*, *R. pseudosolanacearum*, and *R. syzygii*. The *Ralstonia* genus demonstrates diverse lifestyles and has a significant impact on agriculture, making it essential to study the evolution of this genus. Gene expression regulation changes can lead to lifestyle adaptations, and translation efficiency plays a role in this process. The translation elongation is the most prolonged stage of translation, and it highly impacts overall translation efficiency [6,7,8]. As it has been shown in *Escherichia coli*, at least 12% of protein abundance is determined by effectors of translation elongation, while translational initiation determines about 1% of protein abundance, whereas mRNA abundance determines about 53% of protein abundance [9]. However, while mRNA levels strongly depend on the conditions under which genes are expressed, to date, gene-specific regulation of translation elongation efficiency has been identified for only a limited number of genes as a response to stress conditions [10,11], which makes it less volatile and more conservative characteristic with alterations in the course of the evolution of species. Thus, in this study, we focus on the evolution of translation elongation efficiency characteristics in species of the *Ralstonia* genus.

The efficiency of the translation elongation stage depends on mRNA content and structure. The first factor impacting the translation elongation rate is codon bias: there are differences in the frequency of synonymous codons [12,13,14] that correlate with tRNA abundance [15,16,17,18]. Rare codons with the lower-abundance corresponding tRNAs are decoded more slowly than more abundant codons due to the increased time required for the entrance of the conformable tRNA into the ribosome’s A site [7,15,16,17,18]. As a result, genes enriched by common codons are translated significantly faster than genes enriched by rare codons, and therefore, codon bias is associated with translation elongation efficiency [6,7,8] and gene expression levels [6,7,15,19,20,21,22,23]. However, there are known cases when translational pauses induced by rare codons play a functional role in protein folding [24]. Therefore, regulation of translation efficiency stretches beyond codon content optimization.

Another factor that impacts the translation elongation rate is secondary structures in the mRNA. It has been demonstrated that translation occurs through successive translocation-and-pause cycles. Secondary structures in mRNA increase pause duration, which leads to a decrease in the elongation translation rate [25,26,27]. Despite this fact, the abundance and energy of mRNA secondary structures tend to be higher over coding regions compared to untranslated regions [28] due to performing various functions [29], including positive regulation of mRNA half-life, which means that mRNA abundance is correlated with mRNA folding energy [30,31,32,33]. In summary, there is a balance between mRNA folding energy and the codon usage bias that serves to optimize both mRNA [34] and protein [35] expression levels.

The strategy of multi-objective evolutionary optimization of mRNA by codon content, abundance of secondary structures, and energy of secondary structures is often complicated. That is the reason why an organism could have a preferred maintenance strategy of mRNA optimization [36,37,38], which might be associated with various factors affecting the organism, such as the environment, lifestyle, and others [38,39]. However, the questions of how the strategy of translation elongation optimization corresponds to these factors and whether this trait is phylogenetically conserved remain open.

### 1.1. Ecological Diversity of Ralstonia

#### 1.1.1. Soil Bacteria

A group of bacteria commonly found in moist environments such as soil and water [40], which includes species *R. mannitolilytica*, *R. insidiosa*, and *R. pickettii*, represents one of the two branches of the *Ralstonia* genus. As it has been shown for *R. pickettii*, it can survive in nutrient-poor environments and in areas with high metal (copper, nickel, iron, and/or zinc) contamination. It has also been demonstrated that *R. pickettii* strains are capable of degrading/detoxifying several toxic substances, including aromatic hydrocarbons and chlorinated phenolic compounds [41]. As it has been found on plastic-water piping, *R. pickettii* is capable of forming biofilms [42].

The *R. mannitolilytica*, *R. insidiosa,* and *R. pickettii* species also act as opportunistic human pathogens, in particular, as the pathogens responsible for nosocomial infections in immunocompromised patients. The spread of pathogens is provided by the use of contaminated medical solutions [43], including saline, sterile water, as well as disinfectants, and medical equipment [44]. *R. pickettii* is supposed to be the most clinically important pathogen of this group [45,46] forasmuch that most *Ralstonia* human infection reports describe cases related to this pathogen. The ability of *R. pickettii* to survive in disinfectants and pass through filters, which are used to filter-sterilize medical solutions [47], is probably the reason for the persistence of these bacteria in sterile solutions. There have been case reports of meningitis [45,48], infective endocarditis [49,50], nosocomial pneumonia [51], and central line-associated bloodstream infection [4,49] caused by *R. pickettii*. Moreover, there are a few described cases of nosocomial bloodstream infection caused by *R. insidiosa* [4], infective endocarditis [50], and bacteremia [52,53] caused by *R. mannitolilytica* [54]. The infections caused by this group of *Ralstonia* mainly affect people who are immunocompromised, diagnosed with cystic fibrosis, have central venous catheters, or have had recent surgical or medical hospitalizations [49].

#### 1.1.2. Phytopathogens

The second group of *Ralstonia* is plant pathogenic bacteria with a broad host range, including such agricultural plants as potato, tomato, eggplant, and ginger, that cause the widespread disease known as bacterial wilt. Until recently, most Ralstonia phytopathogenic strains were included in *R. solanacearum* species that contained four phylotypes. The differentiation was based on the sequence analysis of the 16S–26S internal transcribed spacer (ITS) region, the endoglucanase gene, or the *hrpB* genes. Moreover, every phylotype had its own area of geographical origin [5]. A taxonomic revision has been proposed to divide the *R. solanacearum* species complex into three species: *R. solanacearum*, *R. pseudosolanacearum*, and *R. syzygii* [55] based on the DNA–DNA relatedness values. *R. solanacearum* and *R. pseudosolanacearum* are pathogens that infect a plant through its roots, affecting a large number of species belonging to more than 50 botanical families [1].

A broad geographical distribution, persistence, and lethality lead to the characterization of these pathogens as the world’s most important phytopathogenic bacteria [56]. There are many important crops among the infected species, including potato, tomato, banana, pepper, eggplant, and tobacco [55]. Unlike other phytopathogenic *Ralstonia R. syzygii* has a narrow host range; the pathogen is vectored by cercopoid insects into clove trees. Banana blood disease strains that have also been classified into *R. syzygii* species infect bananas and plantains via pollinating insects [57].

There are several factors of *Ralstonia* pathogenicity that have been investigated in *R. solanacearum*. The main pathogenicity determinant in *R. solanacearum* is the type III secretion system (T3SS)—a structure that injects type III effector proteins (T3E) into the plant cell cytosol to favor infection. The other factor is exopolysaccharide (EPS), which is important for plant colonization, leading to the occlusion of the xylem vessels that eventually cause the plant wilting symptoms. Other factors impacting pathogenicity are type II-secreted plant cell wall-degrading enzymes responsible for twitching motility (type 4 pili) and swimming motility (flagella) appendages, aerotaxis transducers, cellulases, and pectinases [5]. *R. solanacearum* can survive for years in moist soils or water microcosms until it finds a susceptible host, which means land infection for several years [56].

#### 1.1.3. Cupriavidus Necator: An Outgroup

The species *Cupriavidus necator* is a soil bacteria previously attributed to the *Ralstonia* species and known as *Ralstonia eutropha*. It is known for its resistance against heavy metals and for being another soil bacteria [58] and the closest relative of *R. solanacearum* outside the *Ralstonia* species. These facts make it a perfect candidate to be selected as an outgroup for the *Ralstonia* genus. As it has been shown for the H16 strain of *C. necator*, it is a metabolically versatile bacteria capable of subsisting, in the absence of organic growth substrates, on H_2_ and CO_2_ as its sole sources of energy and carbon [59].

### 1.2. The Genome Structure of Bacteria Belonging to Ralstonia Genus

The genome sequence of the listed bacteria comprises two chromosomes [2]. The size of chromosome 1 of *Ralstonia* genus representatives is about 3.5–4 Mb, and the size of chromosome 2, which is also called megaplasmid, is about 1.4–2 Mb. The size of *C. necator*’s chromosome 1 is about 3.8–4.3 Mb, the size of chromosome 2 is about 2.7–3.4. Additionally, the genome of *C. necator* contains one or two plasmids, whose size is mainly about 0.5 Mb, but some representatives contain plasmids with sizes up to 1.5 Mb. As it has been shown for *R. solanacearum,* both chromosomes have very similar distribution and composition of simple sequence repeats and the presence of compositional biases that indicate joint evolutionary history among the chromosomes [60].

Most housekeeping genes are located on chromosome 1, while chromosome 2 contains chromosome islands and strain-specific genes. Chromosome 2 carries many genes associated with specific lifestyles and pathogenicity, including the Type III and Type VI protein secretion systems, the extracellular polysaccharide (EPS) biosynthesis cluster, flagellar motility determinants, and chemotaxis genes [56]. As it has been shown on *R. solanacearum* before taxonomic revision, the level of synteny of chromosome 1 (70% to 80% of the coding sequences) is higher than the level of chromosome 2 (55–65%). Chromosome 2 is less conservative; 63% of genes are not conserved. The size of chromosome 2 highly varies (26%) among strains compared to the variability of chromosome 1 size (6%) [56].

To sum up, understanding how translation elongation efficiency has evolved in the *Ralstonia* genus can help to broaden the understanding of how these bacteria have adapted to different lifestyles. Such factors as codon bias and mRNA secondary structures play a critical role in translation elongation, and optimizing these factors is necessary for efficient gene expression. However, the strategy for mRNA optimization may vary depending on environmental factors and bacterial lifestyles. This study aims to shed light on the interplay between translation elongation optimization, bacterial lifestyles, and evolutionary divergence in the *Ralstonia* genus. By analyzing different strains within this genus and *C. necator* as an outgroup, we hope to gain insights into the evolution of these bacteria in terms of translation elongation efficiency and its role in their impact on agriculture and human health.

## 2. Materials and Methods

### 2.1. Genome Assemblies Acquisition and Refinement of Their Taxonomical Identity

We selected only complete genomes of *Ralstonia* genus representatives and *C. necator* species that were available in the RefSeq database on 28.01.2021; the number of genomes is 116. The entire set of the genomes’ accessions can be found in Appendix A. Due to the recent revision of the *R. solanacearum* taxonomy, taxonomical identification was further refined for the strains described as *R. solanacearum* using the sequences of the 16S–23S rRNA intergenic spacer (ITS) region, which is utilized for *Ralstonia* phylotypes recognition [55].

### 2.2. Elongation Efficiency Calculation

To employ translation elongation efficiency as a conservative characteristic in the analysis of the evolution of the *Ralstonia* genus, we used Elongation Efficiency (EloE) software v.0.1. The algorithm has been described in the previous studies [36,37,61] and represented in a detail in a recent article [38], we used the same version of EloE software, which is available in the supplementary materials of that study. Briefly, EloE calculates translation elongation efficiency indices (EEI) considering a number of factors in various combinations, namely, the impact of codon content, number of secondary structures, and energy of these structures in mRNA for all protein-coding genes of the studied organism [62]. Various combinations form the five types of translation elongation efficiency indices (see Table 1).

EloE associates a genome under analysis with one of the five types of highly expressed genes’ elongation efficiency optimization, corresponding to these indices based on the position of ribosomal protein genes in the sorted list of EEI values. To determine the elongation efficiency type, the EloE algorithm takes the genome annotation of an organism in the GenBank format as an input and calculates the translation elongation efficiency indices of all protein-coding genes. These genes are sorted by EEI values in descending order, so every gene has its EEI rank. Then, for each EEI type, such characteristics as mean (M) and standard deviation (R) of the ranks of ribosomal protein genes are calculated. A type with the maximum M and, if there are several types with the maximum M, then the one with the minimum R value is defined as an organism-specific type [36,37], also called the base type and the corresponding EEI is called the base index accordingly. We will also call it the translation elongation optimization type further because while varying amongst different taxa, it reflects the impact of major factors affecting their global optimization of gene translation.

The selected type indicates which parameter(s) of the mRNA structure (codon bias, number of secondary structures, energy of secondary structures) highly impact the elongation translation efficiency of the organism under study. The base type EEI values estimate the basal translation elongation rate in the absence of any additional regulation, which can also be considered as a proxy of basal expression of corresponding genes. Thus, it allows us to distinguish a fraction of potentially highly expressed genes (PHEGs) under average environmental conditions.

It is known that complementary subsequences in mRNA tend to fold and form hairpins [36]. EloE calculates local complementary indices (LCI) [37] for each gene, which reflect the average amount and energy of mRNA secondary structures per gene. The LCI means the average stability of a hairpin, which can be formed by local perfect inverted repeats with a certain length mRNA. The LCI1 is calculated by counting nucleotides from complementary subsequences in a gene, and the LCI2 is calculated by counting energy for nucleotides from complementary subsequences in a gene. Thus, LCI1 and LCI2 characteristics correspond to the number and energy of complementary nucleotides per one nucleotide of the analyzed sequence, respectively.

### 2.3. Potentially Highly Expressed Genes Analysis

In this study, potentially highly expressed genes (PHEGs) are defined as a fraction of genes with a high base elongation efficiency index (EEI) value. Based on the distribution of the EEI values for each analyzed strain, a 10% threshold was accepted as distinguishing the fraction of potentially highly expressed genes. The set of PHEG was obtained for each organism under study subsequently.

To compare the average amount and energy of mRNA secondary structures of potentially highly expressed genes across the *Ralstonia* phytopathogenic group, the soil *Ralstonia* group, and the outgroup *C. necator*, joint potentially highly expressed genes were obtained for these groups. These are genes that are present in the PHEG list for at least 80% of the analyzed genomes belonging to each of the groups. Then, for each gene of the resulting list of genes for each genome, the LCI1 and LCI2 values were derived, so the sets of LCI1 and LCI2 values were obtained for each phylogenetic group. Then, the significance of the differences between the LCI values for each group’s potentially highly expressed genes was examined by pairwise comparisons of the LCI values between the groups using the Mann–Whitney U test from the SciPy package [63] using Python.

### 2.4. The Functional Analysis of Potentially Highly Expressed Genes: Clusters of Orthologous Groups (COGs)

Since the list of potentially highly expressed genes (PHEGs) varies according to the selection of base index, the functional analysis of PHEGs becomes an important step to verify that the base EEI reflects the expression level correctly and that the PHEG list does contain genes involved in vital processes of the corresponding groups of microorganisms. Another purpose is to analyze how differences in translation elongation optimization type between phylogenetic groups correspond with their lifestyles. For this, we decided to compare functional groups of PHEGs across phylogenetic groups. To classify the PHEGs by certain functional groups, we used COG (Clusters of Orthologous Groups) identifiers from the COG database [64,65,66]. Each genome was reannotated using the prokka [67] tool, which assigns COG identifiers for the annotated genes, and then the fraction of potentially highly expressed genes (10% of each genome) with corresponding COGs was obtained.

To determine overrepresented COGs in the PHEG set for each phylogenetic group, the number of COGs presented in the phytopathogenic *Ralstonia*, soil *Ralstonia*, and *C. necator* was calculated both for the entire genome and the PHEG list only. Thus, for each COG, three pairs of two-sided exact Fisher’s test were calculated, assessing the pairwise difference in the presence of COGs in the entire gene set and in the PHEG set for the analyzed groups of microorganisms. The confidence level was accepted as sufficient for *p* < 0.017 after the Bonferroni correction for multiple (3) hypotheses.

### 2.5. Analysis of Translation Elongation Characteristics of Orthologous Genes

The translation elongation efficiency characteristics of orthologous genes have been compared to determine how differences in elongation efficiency types reflect the changes in particular genes’ elongation efficiency. The orthologous genes have been defined by the Orthofinder tool [68] for the entire set of protein-coding genes from the analyzed genomes (116). It determined 13,487 orthogroups. While designing a sample of highly represented orthogroups, only those orthogroups that contained at least one gene from each genome were further analyzed, which amounted to 2058 orthogroups.

The translation elongation efficiency characteristics for genes classified to the orthogroups were extracted from the EloE output for each genome. These characteristics include the codon utilization index of efficacy Tai (represents codon usage bias; the higher the Tai value, the less it is enriched in common codons), LCI1 (number of secondary structures in mRNA), and LCI2 (the energy of secondary structures in mRNA). Formulae for each of the characteristics have been described in articles [36,37,38,61].

To assess the similarities and differences in the translation elongation efficiency characteristics between orthogroups of the three analyzed phylogenetic branches, the data about genomes were merged into the characteristics of the phylogenetic branches. For each of the characteristics (Tai, LCI1, and LCI2), the pairwise comparisons between the three phylogenetic groups in each orthogroup were performed using Welch’s *t*-test. The significance level was accepted as sufficient for *p* < 0.05 after applying the Benjamini–Hochberg Procedure for each of the analyzed characteristics. Based on the significance of differences between the phylogenetic branches, orthogroups were classified by each characteristic into 21 groups, reflecting which had the highest, the lowest, or intermediate values.

### 2.6. Translation Elongation Efficiency and Divergence Index

To assess how elongation efficiency corresponds to the evolution of genes, we calculated the divergence index [69] for the entire set of protein-coding genes (EGS) of several Ralstonia genus species using Orthoweb [70]. The divergence index (DI) is the ratio of the number of nonsynonymous substitutions per nonsynonymous site to the number of synonymous substitutions per synonymous site in protein-coding sequences, which is also known as the Ka/Ks ratio [71,72]. We calculated the DI between orthologous genes for the number of genomes belonging to the Ralstonia species following the methodology implemented in [73]. The obtained DI values for the EGS and the information about the genomes used in the analysis are available in Appendix A.

## 3. Results

### 3.1. The Distribution of Ralstonia Genomes by Translation Elongation Optimization Types

The investigation of how the genomes under study are distributed by base EEI types reflecting translation elongation optimization across the *Ralstonia* genus reveals differentiation of the genomes into EEI4 and EEI5 types according to their phylogenetic group (see Figure 1). The strains of phytopathogenic species (*R. solanacearum*, *R. pseudosolanacearum*, and *R. syzygii*) are mostly classified as representatives of the EEI5 type, which is characterized by mRNA optimization taking into account codon composition and local complementarity with the energies of potential secondary structures. The strains of soil bacteria of the *Ralstonia* genus (*R. insidiosa*, *R. mannitolilytica,* and *R. pickettii*) are mostly classified as representatives of the EEI4 type, which is characterized by mRNA optimized, taking into account codon composition and the number of potential secondary structures ignoring their energies. The genomes of an outgroup, *C. necator,* are classified as representatives of the EEI5 type, similar to the phytopathogenic group, although it is soil bacteria similar to *R. insidiosa*, *R. mannitolilytica,* and *R. pickettii*. Therefore, in this case, lifestyle does not determine the translation elongation optimization type.

Since the elongation efficiency type is determined by the relative elongation efficiency of ribosomal protein genes, which is represented by the M (mean rank of ribosomal protein genes EEI) and R (standard deviation of ribosomal protein genes EEI) features, we compared the average M and R values of each type (see Appendix B
Table A1; full table with M and R values for each genome analyzed is available in the Appendix A). The M and R values of the base EEI type revealed that the genomes of phytopathogenic *Ralstonia* are characterized by a lower efficiency of translation of ribosomal protein genes than that of *C. necator*; in turn, the efficiency of translation elongation of ribosomal protein genes in the *C. necator* is lower than in the soil *Ralstonia*. The M, R values for EEI1 type for phytopathogenic *Ralstonia* and *C. necator* evidence that the impact of codon bias on translation efficiency is weak for these organisms; this thesis is also proved by the similarity between the EEI5, EEI2, and EEI3 values. In contrast, codon bias seems to be a significant feature in the translation efficiency optimization of soil *Ralstonia* species’ highly expressed genes. Finally, while soil *Ralstonia* genomes are mostly classified into the EEI4 efficiency type, the translation elongation efficiency characteristics of their ribosomal protein genes are similar for the EEI4 and EEI5 types. Regarding phytopathogenic *Ralstonia* and *C. necator* genomes, those characteristics indicate the prevalence of the role of the energy of secondary structures over their number in the optimization of translation elongation.

Summing up, *Ralstonia* species divides into two groups by translation elongation efficiency type. The first one is represented by phytopathogenic species. Their translation efficiency is optimized against forming stable secondary structures in highly expressed genes with slight optimization by codon content. The second one is represented by soil species that can also act as opportunistic pathogens. Their translation efficiency is optimized against hairpins in general in combination with pronounced codon usage optimization. The outgroup, represented by soil bacteria *C. necator*, demonstrates characteristics similar to phytopathogenic *Ralstonia*.

### 3.2. Comparative Analysis of Translation Elongation Efficiency of Genes in the Genomes of the Two Ralstonia branches and C. necator

The observed differences in elongation efficiency types of *Ralstonia* species are the result of the differences in the elongation efficiency characteristics of genes. Further analysis has been performed in order to estimate the scale of diversity on the level of genes, with a focus on individual translation elongation efficiency characteristics instead of integrated EEI value.

As postulated above, the observed distribution by EEI types points to the strong selection against any secondary structures (not only stable) on highly expressed genes’ mRNA of soil *Ralstonia* genomes. To validate the above statement, the average number and energy (LCI1 and LCI2, respectively, see [38] for the equations) of mRNA secondary structures per nucleotide for potentially highly expressed genes (PHEGs) were assessed for each phylogenetic branch. Both features have demonstrated significantly lower values for PHEGs of soil *Ralstonia* in comparison to phytopathogenic *Ralstonia* and their outgroup (see Figure 2).

Therefore, we conclude that *R. insidiosa*, *R. mannitolilytica,* and *R. pickettii* have decreased both the amount and energy of mRNA secondary structures for the highly expressed genes. It is possible that such a difference is mediated by the shift in translation elongation optimization type.

To determine whether the differences in translation elongation efficiency between the phylogenetic groups are specific for PHEGs or widely present across the genome, we analyzed the distributions of LCI1, LCI2, and Tai (codon utilization index of efficacy, which reflects codon usage bias) for both the entire set of protein-coding genes (EGS) and ribosomal protein genes (RPG) of each analyzed phylogenetic group, as shown in Figure 3. We also included such data for the *Escherichia coli* genome as a representative of EEI1 type, which is strongly optimized by codon usage bias.

The analysis of the LCI1 and LCI2 distributions (Figure 3a,b) demonstrates that the distributions for the EGS are similar for the phytopathogenic *Ralstonia* and *C. necator*. Meanwhile, the distribution for soil *Ralstonia* is shifted leftwards with lower variance, indicating a lower number and energy of secondary structures in the mRNA of their genes. As for ribosomal protein genes, their LCI1 and LCI2 distributions are similar for each of the listed groups. Thus, the LCI1 and LCI2 values discriminate potentially highly expressed ribosomal protein genes from other genes for phytopathogenic *Ralstonia* and *C. necator* better than for soil *Ralstonia*. Remarkably, the distributions of these values in *E. coli* for both groups of genes are strongly shifted leftwards and are substantially overlapping, demonstrating a reduced number and energy of secondary structures in this genome and a slight role of secondary structures in the optimization of highly expressed genes’ translation efficiency. Similarly, soil *Ralstonia* demonstrates the left-shifted pattern of LCI1 and LCI2 EGS distributions, indicating the differences between this group and phytopathogenic *Ralstonia* and *C. necator*.

On the other hand, the analysis of Tai distribution, which reflects codon usage bias, demonstrates that the EGS distributions for the soil *Ralstonia* are skewed to the right compared to the phytopathogenic *Ralstonia* and *C. necator*. Thus, the mean Tai value for soil *Ralstonia* is higher. Considering the fact that such a distribution is shifted to the right for *E. coli* too, which together with higher variance allows us to discriminate highly optimized groups of genes from poorly optimized ones, we conclude that it points out an important role of codon usage bias in soil *Ralstonia* genomes. To sum up, the soil *Ralstonia* demonstrates a reduction in the number and energy of secondary structures with the growth of the role of codon usage bias in translation efficiency optimization of highly expressed genes.

To analyze if variations in these distributions were due to alterations in characteristics of a few genes or a majority of the genes, we examined the translation elongation efficiency characteristics (LCI1, LCI2, Tai) distribution for each orthogroup of each phylogenetic group. We classified 2058 obtained common orthogroups into 21 distinct types based on the observed differences (refer to Figure 4), revealing the diversity of these characteristics among the phylogenetic groups.

We found that translation elongation efficiency characteristics vary among the orthologous genes of the three analyzed phylogenetic groups (see Figure 4). Only 15.6%, 15.7%, and 33.7% of genes demonstrate similar values for the LCI1, LCI2, and Tai, respectively. Notably, the genes of soil *Ralstonia* rarely demonstrate the highest number (LCI1) or energy (LCI2) of secondary structures (see lines 1–5 in Figure 4) in comparison to the phytopathogenic *Ralstonia* and the outgroup. Inversely, the situation when LCI1 and LCI2 are low for soil *Ralstonia* is frequent in the orthogroups (see lines 8–10 and 13–18 in Figure 4). Thus, about 30% of analyzed soil *Ralstonia* genes have significantly lower LCI values than both the other groups, and about 70% of genes have a significantly lower value in comparison to at least one of the groups, while only 3% of the orthologous genes have LCI values significantly higher than in any other group. It supports the idea of stronger selection against secondary structures in soil *Ralstonia*, not only in a narrow PHEG set but for a substantial part of genes.

While phytopathogenic *Ralstonia* and *C. necator* demonstrate very similar distributions for each of the characteristics analyzed, they also have many orthologous genes that differ by translation elongation efficiency characteristics. Thus, about 36% and 29% of phytopathogenic *Ralstonia* genes demonstrate higher numbers of LCI1 and LCI2 than *C. necator*, respectively, and oppositely, 20% and 23% of their genes are less enriched in secondary structures according to their LCI1 and LCI2 values, respectively. The observed differences are a result of mutations in the coding sequences of the respective genes.

The observed differences in the translation elongation efficiency characteristics of genes are supposed to impact the content of the PHEG set; therefore, it is necessary to assess how the subsequent change in the PHEG list corresponds to an organism’s lifestyle and phylogeny.

### 3.3. The Functional Analysis of Potentially Highly Expressed Genes

The main idea of optimization of translation elongation efficiency is the facilitation of highly expressed gene translation; thus, highly expressed genes should be optimized. Following the idea, adjusting the translation efficiency of particular genes during evolution may indicate changes in gene expression, possibly as a result of the adaptation of an organism to the environment [74]. The changes in characteristics of translation elongation efficiency in those genes that used to be highly optimized might result in phenotype changes. Thus, the sets of potentially highly expressed genes (PHEGs) serve as the most suggestive subsamples for a related functional analysis. Here, we perform such an analysis for the genomes of three phylogenetic branches (soil *Ralstonia*, phytopathogenic *Ralstonia*, *C. necator*) with diverse characteristics of translation elongation efficiency. We provide the COG enrichment analysis for both the PHEG and the entire gene (EGS) sets for each of the analyzed phylogenetic groups. The obtained *p*-values are located in Table A2 in Appendix B, and the fractions of each COG obtained by analyzing the PHEGs and by genome-wide analysis for each phylogenetic group are located in Table A3 in Appendix B.

As expected, the most represented COGs refer to the base cell processes: translation, energy production and conversion, and amino acid metabolism (see Figure 5). The fraction of these processes in the PHEG set is much larger compared to the EGS for any group of analyzed organisms, which corresponds to the fact that the genes maintaining these functions are supposed to be highly expressed. Thus, it confirms the assumption that the set of PHEGs obtained by the evaluation of elongation efficiency covers the essential cell processes. Moreover, the COG content at the genome scale is similar for the soil and the phytopathogenic *Ralstonia* groups. That means that while belonging to different EEI types and demonstrating differences in translation elongation efficiency for many genes, the analyzed genomes keep optimizing similar metabolic processes.

Next, we assessed how often particular COGs fall into the PHEG category for each phylogenetic group (see Appendix A). The analysis of the difference in the presence of particular COGs in the PHEG set of the three phylogenetic groups has revealed 81 COGs with significant differences. Most of these COGs (64) tend to be overrepresented in the genomes of soil *Ralstonia* and *C. necator*. The products of these genes are responsible for translation (21 COGs), carbohydrate metabolism and transport, five COGs related to cell wall construction, and five related to carbohydrate metabolism and transport. There are also five genes involved in ion transport (ferredoxin subunit of nitrite reductase or a ring-hydroxylating dioxygenase, ABC-transport system genes COG1653), and three transcription regulation genes (cold shock protein COG1278, COG2183, sigma subunit COG0568). As for cold shock protein, it is worth noting that this protein has been discovered to elicit an immune response in plants [75]; therefore, its overexpression in phytopathogenic *Ralstonia* should be avoided.

A few (7) COGs responsible for different functions are specific for PHEGs of *C. necator*; 6 COGs specific for PHEG set of phytopathogenic *Ralstonia*, where 3 of those genes play a role in response to the environmental changes: HTH-type transcriptional regulatory protein GabR COG1167, Anaerobic regulatory protein COG0664, Regulatory protein AtoC COG2204 involved into the regulation of motility and chemotaxis [76]. Considering functions of corresponding genes, changes in translation efficiency may serve for adaptation to a phytopathogenic lifestyle.

Interestingly, the potentially highly expressed genes tend to be non-divergent in terms of the Ka/Ks ratio (correlation coefficient with the EEI for the entire set of protein-coding genes of *Ralstonia solanacearum* GMI1000 r = −0.3865 *p*-value = 1.6 × 10^−169^). There are a few exceptions, though most of them have unknown functions, and some genes are involved in pathogenicity and are characterized by modest translation elongation efficiency (see Appendix A).

## 4. Discussion

The investigation of translation elongation efficiency in the *Ralstonia* genus has revealed interesting insights into the evolution of these bacteria. We have found that the *Ralstonia* genus, which had been recently described as consisting of two branches [77], had diverged during its evolution by the optimization type of translation elongation efficiency. While there is an agreement between the phylogenetic group and translation elongation efficiency type for the *Ralstonia* genus, the fact that an outgroup represented by soil bacteria *C. necator* belongs to the same translation elongation efficiency type as phytopathogenic *Ralstonia* instead of soil *Ralstonia* group shows that in this case elongation efficiency, optimization type is not accounted for fitness to a particular lifestyle.

We found that the genes of soil *Ralstonia* have significantly lower number and energy of secondary structures in mRNA compared to both phytopathogenic *Ralstonia* and *C. necator* and not only in the potentially highly expressed genes’ (PHEG) set but also in the entire set of protein-coding genes. It impairs the discrimination of highly expressed genes from other genes by secondary structure characteristics (LCI1, LCI2); however, accounting for secondary structures remains necessary for correctly assessing translation elongation efficiency. While the role of secondary structures decreases, the role of codon frequency content optimization increases for the highly expressed genes in soil *Ralstonia*. Since GC-rich sequences tend to form more stable structures, GC content may also play a role in the observed differences between species, which is supported by the fact that it is lower in soil *Ralstonia* (64%) than in phytopathogenic *Ralstonia* (67%) and *C. necator* (66%). The optimization of translation elongation efficiency of highly expressed genes by codon usage bias is common for fast-growing organisms [78], serving as an adaptation for maximizing the rate of protein production and subsequent division [79]. However, the analyzed microbes demonstrate quite similar minimal doubling time rates: 2.3 h for *R. solanacearum* GMI1000 [80], which represents phytopathogenic *Ralstonia*, and a minimum 1 h (in well-optimized conditions) for *C. necator* [81] and about 2 h for *R. insidiosa* [82], which represents soil *Ralstonia*. Thus, there is no evidence that detected diversity in translation elongation efficiency in the *Ralstonia* genus, serving as an adaptation for raising bacterial growth rate.

The differences in the type of translation elongation efficiency have not been formed by changes in gene repertoire or mutations of particular genes (in particular, ribosomal protein genes). Conversely, the differences in number, energy of secondary structure, and codon usage bias have been detected for most orthologous genes. Notably, phytopathogenic *Ralstonia* and *C. necator* demonstrate many orthologous genes differing by translation elongation efficiency characteristics due to mutations in coding sequences that occurred during evolution, which may impact the translation elongation efficiency of particular genes. Since *C. necator* is an outgroup, we could expect more differences in coding sequences between it and *Ralstonia*. However, more orthologous genes differ between soil *Ralstonia* and phytopathogenic *Ralstonia* than between phytopathogenic *Ralstonia* and *C. necator* by translation elongation efficiency characteristics. It points to driving selection for reducing the number and energy of secondary structures in protein-coding genes of soil *Ralstonia* species.

Such variability in the characteristics of translation elongation efficiency affects the content of the potentially highly expressed genes (PHEG) set, which reflects in the presence or absence of some particular genes in that set. However, most of the genes in the PHEG set are common among analyzed species and are related to the primary metabolism, which speaks in favor of adequate representation of potentially highly expressed genes by respective EEI indices. While the elongation efficiency types distribution of the analyzed species do not correspond to their lifestyles, several genes involved in specific responses to the environment were shown to be overrepresented in the PHEG set of phytopathogenic *Ralstonia*, which points to the possible role of elongation efficiency alterations in particular genes as adaptations to the lifestyle. Overall, these findings shed light on the complex interplay between secondary structures in mRNA and codon usage bias in the optimization of translation elongation efficiency that shapes the evolution of the *Ralstonia* genus.

## 5. Conclusions

In this study, we have found that translation elongation efficiency characteristics diverge in accordance with the phylogeny of the *Ralstonia* genus. Soil *Ralstonia* have a lower number and energy of potential secondary structures in their mRNA and an increased role of codon usage bias in optimizing highly expressed genes’ translation elongation efficiency in comparison to both phytopathogenic *Ralstonia* and an outgroup, represented by soil bacteria. The differences in the characteristics of translation elongation efficiency have been detected for most orthologous genes, affecting the sets of potentially highly expressed genes constructed based on translation efficiency characteristics. We have demonstrated that certain genes involved in specific responses to the environment are overrepresented in the potentially highly expressed genes set of plant pathogenic *Ralstonia*, suggesting that alterations in the elongation efficiency of these genes may have played a role in the adaptation to their lifestyle. Overall, we believe that the presented analysis of the characteristics of translation elongation efficiency has shown to be a promising approach for studying the complex mechanisms that determine the evolution and adaptation of bacteria in various environments.

## Figures and Tables

**Figure 1 biology-12-01338-f001:**
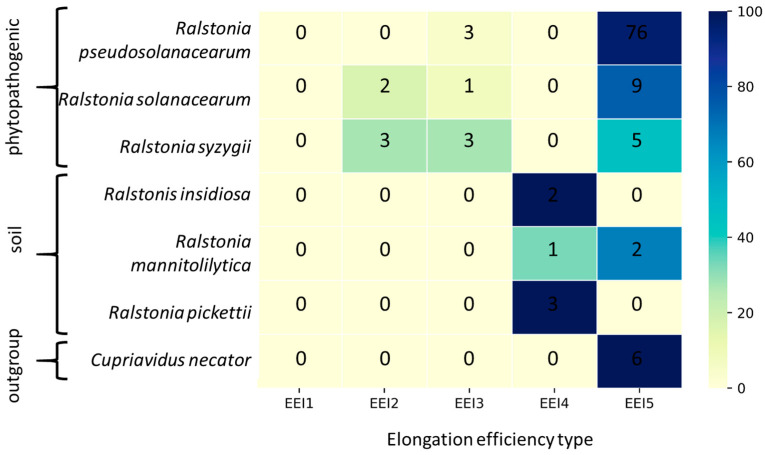
The distribution of the *Ralstonia* species among EEI optimization types. The number of genomes that belong to certain species and are characterized by a certain elongation efficiency optimization type is specified inside corresponding tiles. Color depicts the percentage of genomes belonging to a certain optimization type among all the studied genomes of a corresponding species.

**Figure 2 biology-12-01338-f002:**
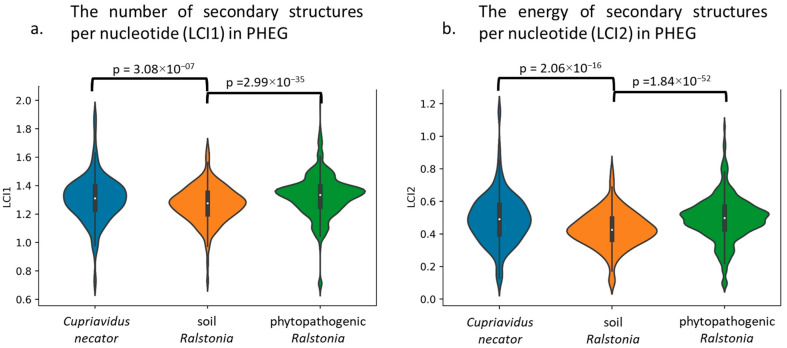
The mean numbers of mRNA secondary structures (LCI1) per nucleotide or energy of secondary structures per nucleotide (LCI2) among potentially highly expressed genes (PHEGs) for *C. necator*, soil *Ralstonia*, and phytopathogenic *Ralstonia*.

**Figure 3 biology-12-01338-f003:**
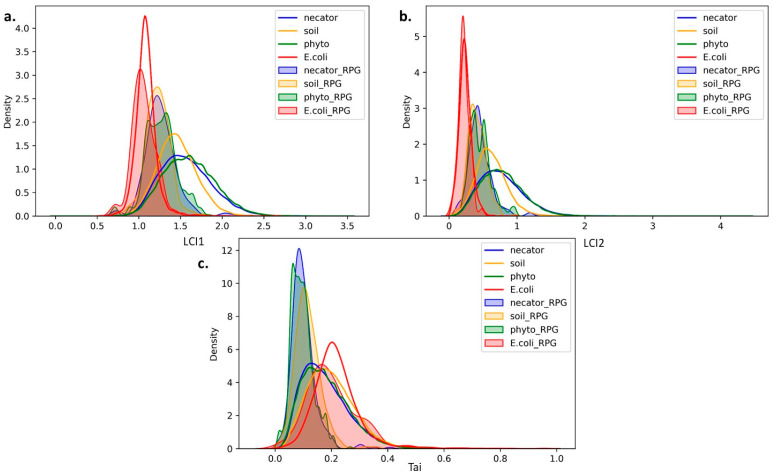
The distributions of both the entire set of protein-coding genes (EGS) and ribosomal protein genes (RPG) for the genomes from the three bacterial phylogenetic groups: soil *Ralstonia*, phytopathogenic *Ralstonia*, and *Cupriavidus necator* and for *Escherichia coli* genome by translation elongation efficiency characteristics: (**a**) mean number of secondary structures per nucleotide (LCI1); (**b**) mean energy of secondary structures per nucleotide (LCI2); (**c**) codon utilization index of efficacy (Tai).

**Figure 4 biology-12-01338-f004:**
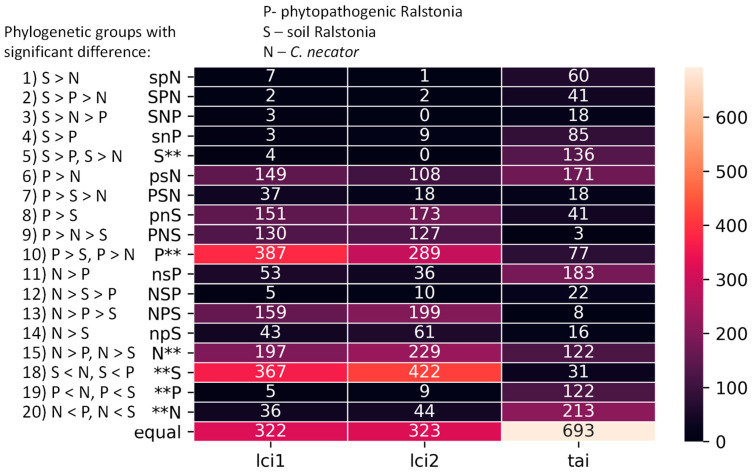
The types of translation elongation efficiency distribution in the orthogroups of phylogenetic groups. The order of letters P (phytopathogenic *Ralstonia*), S (soil *Ralstonia*), or N (*C. necator*) refers to the value of the characteristic analyzed—the leftmost letter has the highest mean value, the rightmost letter has the lowest mean value. Substitution of letters by “**” is used in case of similar values with no significant difference. The size of letters refers to the significance of differences between phylogenetic groups: capital letter demonstrates a significant difference between this phylogenetic group and the next letter. For instance, PSN represents orthogroups with significant differences between each group, where P has the highest value; psN represents orthogroups with similar values for phytopathogenic and soil *Ralstonia* and significantly lower values for *C. necator*. Each group is additionally described in the picture.

**Figure 5 biology-12-01338-f005:**
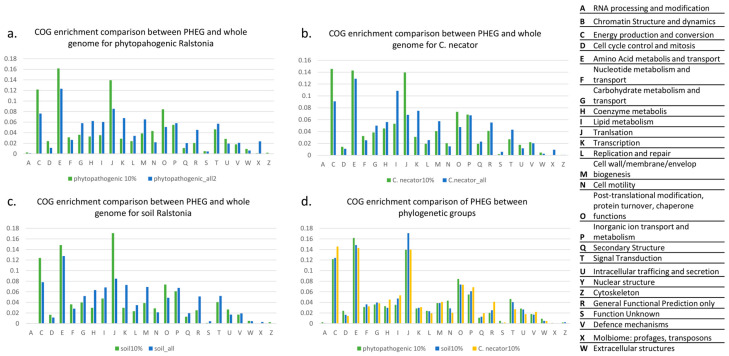
The comparison between the COG (Clusters of Orthologous Groups) enrichment for the PHEGs (10%) and the entire gene sets (all) for genomes belonging to (**a**) phytopathogenic *Ralstonia*, (**b**) *C. necator*, and (**c**) soil *Ralstonia*. (**d**) The comparison of the PHEGs’ COG enrichment for genomes of these phylogenetic groups. The right part of the picture describes the functions of the COGs.

**Table 1 biology-12-01338-t001:** Factors of translation elongation efficiency considered in each type of the translation elongation efficiency indices.

EEI Type	Codon Composition	The Number of mRNA Hairpins	Energy of Potential Secondary Structures in mRNA
EEI1	+		
EEI2		+	
EEI3			+
EEI4	+	+	
EEI5	+		+

## Data Availability

All data generated during this study are included in this published article.

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
