# Peer review of "Bioinformatic Analysis Reveals the Role of Translation Elongation Efficiency Optimisation in the Evolution of Ralstonia Genus"

_biology, 2023, doi:10.3390/biology12101338_

Round 1
Reviewer 1 Report
Manuscript ID: biology-2608259 "Bioinformatic analysis reveals the role of translation elongation efficiency optimisation in the evolution of Ralstonia genus" by Korenskaia and collaborators has been resubmitted to the section of Bioinformatics. The manuscript has gone through peer review in the previous submission in which points raised have been resolved. I have no other issues with the manuscript and will recommend acceptance for publication after review by the editing team.
Minor editing of English language by the editing team will finalize preparation for publication.
Author Response
Dear reviewer,
Thank you for the review!
Reviewer 2 Report
This manuscript is predicated on a premise that different bacterial species fall into different categories in respect to their mode of "translation elongation optimization". The authors believe that there are five different modes of optimization and that we can assign a bacterial strain to one of these five modes by calculating five index values (EE1, EE2, EE3, EE4, EE5) and ascertaining which of the five index values has the maximum value. This is certainly an interesting idea but I am not sure that I am convinced. But for the purposes of this review, let's assume that the premise is correct.
Based on the premise described above, the authors surveyed a set of around 100 complete genome sequences from bacteria of the species Ralstonia. The vast majority of these genomes came from species R. pseudosolanacearum and a few came from each of the other species in the genus. For each genome, the authors calculated the indices and thus assigned each strain to a category to reflect its mode translation elongation optimization. Their main findings are summarized in Figure 1 and are briefly:
(1) Among the three phytopathogenic species, EEI5 was maximal.
(2) Among two soil-associated species, EEI4 was maximal.
(3) Among one of the soil-associated species, EEI5 was maximal (in common with the phytopathogens).
The discussion and conclusions of the paper are mostly based on these three findings. I have concerns about the validity of these findings and thus I have concerns about the authors' conclusions.
The authors' main message is that there is a clear distinction between the phytopathogenic species versus the non phytopathogenic species. That conclusion is unsafe because:
(1) One of the three non-phytopathogens (R. mannitolilytica) has highest value for EEI5, i.e. the same as for the phytopathogens (see Figure 1).
(2) The numbers of genomes sampled for the non-phytopathogens is tiny: just 2, 3 and 3 genomes for each of the three species. The amount of inter-strain variability is high. For example, a third of the R. mannitolilytica strains maximise EEI4 while two thirds maximise EEI5. This suggests that if a larger, representative sample of genomes were investigated from these species, then we might see a different pattern. In other words, what we see here is just stochastic noise rather than biologically meaningful signal.
Now, it is possible that I have completely misunderstood the study. In that case, I would urge the authors to make their manuscript more clear and concise. On the other hand, if I have not misunderstood, then their conclusions are not robustly supported by their data.
The authors' case might be stronger if they could demonstrate that clade-specific specialization of optimization is seen elsewhere in other bacterial taxa, ideally taxa for which a much larger number of diverse strains have been sequenced. It would be extremely surprising that this phenomenon is unique to Ralstonia and I would be more convinced if there is evidence that it exists elsewhere too.
Aside from my major reservations, I have some more minor comments:
Line 33. I don't understand what 'higher role' means. How can a role have height?
Line 39. I disagree with the claim that this approach provides any non-trivial insight into complex mechanisms.
Lines 46-53. This section needs to be supported by references.
Lines 45 - 187. The Introduction section is much too long. It contains too much background, textbook information and it contains quite a lot of repetition. For example compares lines 45-48 with lines 95-99 and lines 132-137.
Line 126. Please remember to use italics for gene symbols, e.g. hrpB.
Lines 130. Is this statement also true for R. syzygi or not? Also, line 139 - 144: do the other two species also use a T3SS or not? Currently the text is a little unclear. But, arguably, a lot of this information could simply be deleted from the text.
Lines 171-174. Which reference supports the claims about % synteny? It might be necessary to add further citation(s) here.
Line 213. What is a 'genome card'?
Line 309. The title of this section mentions "distribution of ... genomes". But the text does not talk about distribution of genomes; it talks of distribution of "base EEI types". Please be consistent between the title and the text.
Lines 322-323. This observation does not exclude the possibility that lifestyle does not determine TEO type. Maybe more than one lifestyle can favour the same TEO type. After all, there are only a few TEO types and yet there are many different lifestyles that a bacterium could adopt. So, inevitably each TEO type will be observed in multiple lifestyles.
Figure 1. It would make much more sense to colour-code according to proportion rather than absolute number.
Lines 362-369. I think the authors are postulating that there might be selection against mRNA secondary structure in the highly expressed genes of soil-associated species of Ralstonia. This seems plausible. But is there any evidence that such selection is absent from other species? Isn't such selection universal? I'd like to see more robust testing of this hypothesis. In the literature, what other analyses and techniques have been used to study selection pressure acting on mRNA secondary structure?
The text is generally intelligible, despite quite a few minor typographical errors of grammar, spelling or punctuation. The minor deficiencies are not a barrier to understanding the authors' intended meaning.
Round 2
Reviewer 2 Report
I thank the authors for their detailed consideration of my previous comments and suggestions. Though I still have some reservations, overall the manuscript is significantly improved and merits publication.
The authors have corrected many of the minor errors and the authors' intended meaning is clear.